# An Improved Two-Step Floating Catchment Area (2SFCA) Method for Measuring Spatial Accessibility to Elderly Care Facilities in Xi’an, China

**DOI:** 10.3390/ijerph191811465

**Published:** 2022-09-12

**Authors:** Linggui Liu, Han Lyu, Yi Zhao, Dian Zhou

**Affiliations:** 1School of Human Settlements and Civil Engineering, Xi’an Jiaotong University, Xi’an 710049, China; 2Technology Innovation Center for Land Engineering and Human Settlements, Shaanxi Land Engineering Construction Group Co., Ltd. and Xi’an Jiaotong University, Xi’an 710049, China; 3School of Humanities and Social Science, Xi’an Jiaotong University, Xi’an 710049, China

**Keywords:** aging, elderly care facilities, accessibility, 2SFCA model, ArcGIS

## Abstract

With the continuous aging of society, the demand among elderly citizens for care facilities is increasing. The accessibility of elderly care facilities is a significant indicator for evaluating whether the layout of urban elderly care facilities is reasonable, and research on the spatial accessibility of related facilities has become an important academic issue in recent years. In this paper, based on the lack of accurate measurement in calculating the spatial accessibility of existing elderly care facilities, we improve the mathematical model based on the two-step floating catchment area method (2SFCA) and introduce the probability function of the elderly population’s choice, taking into account the influence of institutional capacity and service quality. In terms of the catchment radius, the calculation accuracy is improved by using the shortest distance along the route combined with the real road network instead of choosing the Euclidean distance. In addition, specific travel thresholds are set for the travel characteristics of the elderly. An evaluation model of the accessibility of urban elderly care facilities is constructed with the help of ArcGIS software to evaluate and analyze the accessibility of the current layout of urban elderly care facilities in Xi’an, China. The results show that the improved 2SFCA model is more effective in evaluating the spatial accessibility of elderly care facilities and has higher accuracy than the previous calculation model, which provides a methodological basis and academic reference for the specific planning of urban elderly care facilities.

## 1. Introduction

Prominent population growth in urban cities and the aging of society are two trends in cities worldwide, especially those in developing countries [1]. To some degree, population aging is a sign of progress in social development, but it also puts great pressure on social pensions. Nowadays, the aging of the population is a common concern of the international community. The global proportion of the population aged 65 and older rose from 5% in 1960 to 9% in 2018 and is expected to rise to 16% by 2050, when nearly 80% of the elderly population will be living in developing countries [2]. The rapid growth of the elderly population has become a prominent and global phenomenon, including in developed countries such as Canada [3] and Japan [4] and developing countries such as Thailand. Since the 21st century, the aging process in China has accelerated significantly and the problem of elderly care has become increasingly prominent. With the increase in aging and the burden of family retirement, there is a growing demand for elderly care facilities. According to the data of the seventh population census [5], the fertility rate of China’s population continues to decline, the proportion of the elderly population continues to rise, and the age which people reach is increasing. Compared with developed countries, China’s aging population is characterized by a large scale, rapid growth rate, and aging beyond their wealth.

Spatial accessibility is an important reference indicator to determine whether the layout of various facilities is reasonable. Lee and Kim pointed out that the reasonable layout of public health facilities is crucial to their service quality and satisfaction, and is one of the basic needs of residents [6]. Therefore, it is vital to analyze the accessibility of elderly care facilities for elderly residents in different communities. In addition, since the demand for elderly care facilities varies in space and time [7], understanding the variation in demand can help decision makers to better arrange the geographical distribution of facilities and the related service hours. Therefore, scientific accessibility analysis is the basis for judging the variability of regional elderly services and optimizing the spatial allocation of resources. Spatial accessibility includes three basic elements: population, service supply level, and distance. The two most commonly used models for accessibility evaluation are the potential model and the two-step floating catchment area method (2SFCA) model. Xi’an is an important city in the development of Western China. At present, under the active guidance of national policies, elderly care facilities are developing rapidly in terms of quantity and quality, but the problem of an unbalanced supply–demand structure is still prominent. Therefore, the application of scientific methods to evaluate the spatial accessibility of Xi’an elderly care facilities can help researchers to better understand the layout of elderly care facilities and alleviate the problems brought by the aging society. According to Sunwei Liu [8], the results of a comparison of the 2SFCA and other potential models to evaluate community facility accessibility in Beilin District, Xi’an, China, demonstrated that the 2SFCA model was more suitable for short-distance walking scenarios because it represented the actual usage of community care facilities. Because the normal 2SFCA method is not effective enough, we used an improved 2SFCA method to calculate the accessibility of Xi’an. Additionally, this study includes results from a comparison between the improved and the normal methods, which provide a more intuitive reference point for planners. 

## 2. Literature Review: Two-Step Floating Catchment Area (2SFCA) Model

The two-step floating catchment area (2SFCA) model measures spatial accessibility in two steps. First, it determines the number of people in the neighborhoods near each elderly care facility to calculate the supply–demand ratio within the search radius. Then, for each residential area, it searches all elderly care facilities within the threshold travel time and determines the supply–demand ratio to derive the accessibility value of the residential area. The 2SFCA method is widely used for the evaluation of accessibility in public service facilities such as healthcare services [9], schools, elderly care facilities, and green spaces, where the green-space-accessibility-related studies have become a research hotspot in recent years [10], and the Gaussian 2SFCA method is generally used.

The previous literature shows that improvements in the 2SFCA model in recent years have focused on four aspects: the first aspect is to extend the model by adding spatial decay weights; Luo and Qi developed an enhanced 2SFCA (E2SFCA) model [11]. This model divides travel time into discrete time periods and assigns different weights to the different time periods. Other researchers use continuous functions such as kernel density [12] or Gaussian function to calculate the weights. The second aspect is to adjust the search radius based on specific accessibility factors, such as population density and facility level [13]. For example, if the population density is low in some areas of the study area and the conventional search radius cannot cover the effective residential areas, then the search radius is reset. The third area of focus is to extend the supply-and-demand model embedded in 2SFCA [14], which takes into account the competition between facilities and considers that the demand for care in one facility is influenced by the nearby facilities, thus affecting the specific choice of the person. Combining the Huff model with the 2SFCA model (Luo, 2014) [15], a modified 2SFCA model was proposed [16], which calculates the paired supply ratio for each population unit and hospital. These models add choice weights to each healthcare facility to simulate the population’s preferences. The fourth aspect focuses on the travel behavior from the location of the population to the corresponding supply point location. Langford [17] argued that patients using different modes of transportation should have different travel time thresholds, and they modified the 2SFCA model to calculate the separate supply–demand ratios for different modes of transportation.

Although the model has been continuously improved by scholars over the years, most studies still analyze the accessibility of the facilities in terms of administrative areas. These studies mainly use census data to represent the number of inhabitants, assuming that the population of an area is evenly distributed and remains constant over time; meanwhile, in order to simplify the model, most studies do not take factors of the actual road network characteristics into account, which means that the entire supply–demand relationship is static and limited to the residential area of each potential demand object. This leads to the problem of uncertain geography [18], resulting in an inaccurate measurement of spatial accessibility [19]. This issue is particularly important for analyzing the demand for sensitive services in medical-type elderly care facilities and other similar contexts. When more urgent services are needed, the elderly population needs to be served from a geographic setting in their current location, as characterized by a specific road network. Therefore, assuming that all groups of the elderly population in the study area are potential service users, we expect the relevant institutional accessibility values to vary according to different road network characteristics, taking into account the factor that the elderly population’s mode of travel is primarily walking. We adjusted the catchment radius thresholds based on this characteristic. In the neighborhood unit theory, created by American sociologist Clarence Perry in 1929 when he developed the New York Regional Plan, the ideal radius of a neighborhood is between 400 m and 800 m, where 800 m corresponds to about a 10 min walk [20]. Considering that elderly people can be slower than healthy younger people in terms of walking speed, the study by Padeiro [21] concluded that the maximum acceptable time it should take for an elderly person to walk to the nearest pharmacy is 15 min by asking 30 elderly people; at an average speed of 0.8 m/s, an elderly person will walk 720 m in 15 min, which is less than 800 m. Based on the above neighborhood unit theory and the walking range of the elderly population, the mathematical model in this paper sets 800 m as the threshold value.

The main goal of this study is to improve the model for measuring the accessibility of elderly care facilities by introducing a probability function of elderly people’s choice in the model, taking into account the influence brought by institutional capacity and service quality. Instead of choosing a Euclidean distance for the search radius, the shortest distance along the route, as derived by using the real road network, is used. Compared with other medical facilities, the service radius and characteristics of elderly care facilities are different; thus, specific travel thresholds are set for the travel characteristics of the elderly, and an evaluation model for the accessibility of urban elderly care facilities is constructed with the help of ArcGIS software to evaluate and analyze the accessibility of the current layout of urban elderly care facilities. This study includes the comprehensive results of different quantitative methods to evaluate the accessibility of the selected areas in a hierarchical manner, hoping to provide methodological references for relevant academic research and more intuitive decision-making references for city managers and planners.

## 3. Data and Method

### 3.1. Research Area

The study area of this paper was the main urban area of Xi’an, which consists of six administrative divisions, including Xincheng District, Beilin District, Lianhu District, Baqiao District, Weiyang District, and Yanta District, and contains 53 street offices with an area of 843.56 square kilometers (Figure 1). According to the data of the seventh census of Xi’an published by the Xi’an Bureau of Statistics, the population of Xi’an aged 60 and above was 207,518, accounting for 16.02% of the city’s population, of which 14,117,727 were aged 65 and above, accounting for 10.90% of the city’s population. Compared with the sixth national census, the proportion of the population aged 60 and above increased by 3.48 percentage points, and the proportion of the population aged 65 and above increased by 2.44 percentage points. It is thus concluded that the proportion of the elderly population in Xi’an is increasing year by year, and the aging rate is steadily increasing. Among them, 14.16% of the population aged 65 and above reside in Beilin District, 13.66% reside in Xincheng District, 12.06% reside in Lianhu District, 22.79% reside in Yanta District, 10.15% reside in Baqiao District, and 8.15% reside in Weiyang District [22]. In addition, the main urban area is located in the Guanzhong Plain, with concentrated building construction, dense roads at all levels, easy travel for the elderly, and a large potential demand for elderly care facilities.

### 3.2. Data Source

The spatial data used in this paper include open street maps (OSM), POI data, and road network data, etc. The demographic data include the data of the 7th population census and the data of the elderly population in 2020, as provided by the Civil Affairs Bureau of Xi’an City and each street office. The type and scope of the data can meet the needs of the study, and the combination of spatial data and demographic data makes the results more valid and reliable. POI is an abbreviation of “Point of Interest”, which is a spatial entity in GIS and can visually reflect the distribution of urban public service facilities. This paper integrates the above data to study the convenience of elderly people’s life in six urban areas of Xi’an from the street scale (Figure 2).

#### 3.2.1. Network Data

The road network data used in this paper include road network data from OSM (OpenStreetMap) at different levels, such as main roads, secondary roads, and feeder roads. The road network is established by using the road centerline layers of six administrative districts in Xi’an, and the path distance between any two points in the city is measured according to the actual road network. In terms of data processing, the “Street_Centerlines” layer contains information on the basic road attributes. Among them, the field “Directiona” stores the road direction information, which contains four values: “Two Way” represents a two-way road; “One Way (Digitizing direction)” indicates a one-way road along the direction of the line element, i.e., the direction goes from the beginning to the end of the line element; “One way (Against digitizing direction)” indicates a one-way street with a direction opposite to the digitizing direction of the line element, i.e., the direction is from the end to the beginning of the line element; for the element with some “Unknown” values, the direction is uncertain, so these are assumed to be two-way roads. In the path analysis, the “Directiona” field has an important role. Therefore, the one-way streets in the layer are filtered by this field, and the appropriate arrow style is set to show the traffic direction of the street.

#### 3.2.2. Population Data

The people affected by the accessibility of elderly care facilities are the elderly; therefore, this study collated data for the elderly population using python to calculate the number of households in residential areas in six urban areas of Xi’an. After removing duplicate values and values which were invalid due to demolition, 856 residential areas were obtained, covering 53 street office divisions, and the corresponding residential elderly populations were derived by combining the aging rates in the records in different street offices. ArcGIS was used to determine the ratio between the elderly population number and the area of each street to derive the elderly population density level distribution, which was precisely positioned on the map of six urban areas (Figure 3). The results show that there is a large difference in population density, with the highest value, such as that on Taiyi street, reaching 8571.945 persons/square kilometer, while the lowest value in the northern peripheral streets is only 58.188 persons/square kilometer. In terms of spatial distribution, the degree of aging in Xi’an shows a circling structure of central concentration, inner edge diffusion, and outer edge increment.

#### 3.2.3. Elderly Care Facilities Data

We obtained information on a total of 123 elderly care institutions in Xi’an from the official government website of the Shaanxi Civil Affairs Department, which reduced to 53 elderly care institutions after removing those outside the main urban area. The data included information on the geographical coordinates, specific names, bed capacities, and economic standards of the elderly care institutions. We used bed capacity as the weight and the size of poi points in ArcGIS to indicate the specific distribution of elderly care institutions (Figure 4). In this study, we used the “spatial autocorrelation” tool in ArcGIS 10.2 to analyze the spatial clustering degree of the elderly care institutions, and we obtained the results of the clustering degree of the elderly care institutions in Xi’an (Figure 5). The results showed that the nearest neighbor index was less than 1, the z-value was 3.263, and the score was more than 2.58. The *p*-value was 0.001: less than 0.1. The distribution of elderly care institutions showed a “pattern of dense core and sparse periphery”. The central area of the city where Beilin District, Lianhu District, and Xincheng District are located was found to be the area with the highest density of elderly care facility distribution, and a few high-density clusters were found at the southwest and northwest edges of the city. 

### 3.3. An Improved Two-Step Floating Catchment Area

The 2SFCA model is a simplification of the classical gravity model [23], which is widely used to measure the accessibility of public facilities and urban green spaces [24]. The two-step floating catchment area method is used to compare the number of facilities that are accessible to residents within a given threshold by setting a critical value of travel distance, and the larger the value the better the accessibility. The advantage of this method is that it takes the spatial distance factor into account while incorporating the distribution of resources into the accessibility evaluation.

First, we found a distance-decay law between elderly care facilities and residential areas in the region [25]. Therefore, the traditional 2SFCA model was improved by using a distance-decay function instead of the dichotomous method to deal with distance decay. Three continuous distance-decay functions (power function, Gaussian function, and kernel density function) were considered a result of the uncertainty in the distance-decay law for the behavior of inhabitants within the radius of a certain watershed at the regional scale. According to previous studies, the Gaussian function is more suitable, and its expression is [26]:(1)$$\begin{array}{c} f\left(d_{ij},d_{0}\right)=\frac{e^{-\left(1/2\right)\times {\left(d_{ij}/d_{0}\right)}^{2}}-e^{-\left(1/2\right)}}{1-e^{-\left(1/2\right)}} \end{array}$$

After adding the continuous distance-decay function, the following steps and expressions of the 2SFCA method were used:

Step 1: For an elderly care facility, *j*, search for all populations, *k*, within a threshold travel distance, *d*_0_, from facility *j* and calculate the supply–population ratio, *R_j_*, within the search area:(2)$$R_{j}=\frac{S_{j}}{\sum_{k\in \left[d_{kj}\leq d_{0}\right]}P_{k}f\left(d_{kj}\right)}$$
where *P_k_* is the population whose center of mass at location *k* falls within search area *j*; *S_j_* is the capacity of the elderly living facility *j*; *d_kj_* is the travel distance from *k* to *j*. Since the traditional 2SFCA model ignores the competition among elderly care facilities influenced by service quality, area, and location, and residents prefer high-quality facilities or facilities closer to their homes, the size of the search radius was used to represent the service provision level of a certain facility in the traditional 2SFCA model. In addition, the level of elderly care services is not only related to its size but also to the quality of services. In general, the higher the level of a business is, the higher the level of service will be. Therefore, the level (quality indicator) and competition among elderly care facilities are considered in the improved model. Specifically, the Huff model was used to calculate the probability of selecting an elderly care facility for each township/street population. The expression is:(3)$$Prob_{ij}=\frac{S_{j}f\left(d_{ij}\right)}{\sum_{k\in \left[d_{ik}\leq d_{0}\right]}S_{k}f\left(d_{ik}\right)}$$
where *prob_ij_* is the probability of *i* choosing *j*; *S_j_* is the level of elderly care facility supply, determined by the combination of area and rank; and *f*(*d_ik_*) is the decay function of the distance between *i* and *k*. The introduction of the Huff model in 2SFCA takes both the time cost between residential points and elderly care facilities and the attractiveness of the facilities to the elderly into account, thus incorporating the choice behavior of residents into the accessibility calculation. In addition, the scale of supply was changed from the scale of supply to the level of supply (S) to reflect the service capacity of elderly care facilities. Finally, based on the formula, the accessibility of elderly care facilities was calculated using the improved 2SFCA model proposed by Delamater based on 3SFCA. The model absorbs the advantages of previous studies and takes the influence of the road network characteristics on the accuracy of accessibility measurement into account; the calculation process is: 

Step 1, for each elderly care facility:(4)$$R_{j}=\frac{S_{j}}{\sum_{k\in \left[d_{kj}\leq d_{0}\right]}P_{k}f\left(d_{kj}\right)Prob_{kj}}$$

Step 2, for each township (or street) (*i*): (5)$$A_{i}^{F}=\underset{j\in \left[d_{ij}\leq d_{0}\right]}{\sum }Prob_{ij}f\left(d_{ij}\right)\frac{S_{j}f\left(d_{ij}\right)}{\sum_{k\in \left[d_{kj}\leq d_{0}\right]}P_{k}f\left(d_{kj}\right)Prob_{kj}}$$

### 3.4. Catchment Radius

Measuring the distance to elderly care facilities is crucial for calculating the accessibility value. Urban networks play an important role in the formation of urban activities. The network consists of four components: link, node, center, and stop. In network analysis, roads and other linear entities are topologized as “lines”. The spatial accessibility of space is analyzed by topologizing the starting point and the target point as points, and then constructing the spatial attribute characteristics of the entity objects, simulating the flow and distribution of resources in the network [27]. In terms of search radius, the model does not choose a flat Euclidean distance measure as the search radius but uses the shortest path distance along the network through network analysis in a GIS environment, combined with the OD cost matrix, to make the calculation results more accurate and reasonable. Network analysis is based on graph theory and operation research to model and geographic networks and urban infrastructure networks in order to find the shortest paths and obtain the best resource-allocation method. The principle of the model search radius is shown in Figure 6.

In this study, the center represents the elderly care institutions, the nodes represent each intersection, and the impedance represents the time spent walking on each roadway. In this way, each nursing home is a collection of centers, and the topology is constructed based on road paths to build a road network dataset. The service area of each center is calculated by the “new service area” function in ArcMap network analysis and is used as the service area of the elderly care institutions. Generally speaking, elderly people travel mainly by walking, and this paper selects the walking mode of elderly people for distance calculation. Therefore, in the road network dataset, the service area of 15 min’ walk from each elderly care facility was calculated and used as the search radius for this study. On the ArcGIS platform, the road network data of the main urban area of Xi’an is used to establish the urban road database, perform topology, and build the road dataset.

## 4. Results 

### 4.1. Case Studying

Figure 7 shows the results of accessibility of elderly care institutions in six urban areas of Xi’an. The accessibility status of street offices was classified into 10 levels. It can be seen from the figure that, when the search radius is 800 m, the accessibility distribution in the main urban area of Xi’an is characterized by differences; the lowest value is 0.000 and the highest value is 1.291, from which it can be seen that the spatial accessibility distribution is not uniform. Among them, the high accessibility values are mainly concentrated in Yuhuazhai Street, Zhanbagou Street, Electronic City Street, and Qujiang Street. In general, the high accessibility values at the street office level in the main urban area of Xi’an are concentrated in the peripheral southern Yanta area. Although the density of institutions in Yanta is not as high as that in the central urban area where the Beilin, Xincheng, and Lianhu Districts are located, the population density in Beilin is high, so Yanta has more resources per capita within the service radius of the street office level elderly care institutions, which leads to the concentration of high accessibility values in Yanta.

Xincheng District, Beilin District, and Lianhu District are the old urban areas of Xi’an, and are the political, economic, and cultural centers of Xi’an. Among them, the service scope and service types undertaken by the elderly care institutions facilities in Qingnian Road Street, Jiefangmen Street, Beiyuanmen Street, and Taiyi Road Street are obviously higher than those of some elderly care institutions in the peripheral areas, which require higher-quality service facilities and the corresponding highly trained staff, and their location choices clearly reflect the economic orientation of the residents. The old city has highly developed economic conditions, refined infrastructure, and convenient transportation, and these factors ensure the normal operation of elderly care institutions; therefore, a large number of elderly care institutions are laid out in this area, making it a high-value area for the accessibility of elderly care institutions.

The accessibility values of elderly care institutions in Yanta, Baqiao, and Weiyang Districts are different, with Yanta and Weiyang Districts having higher accessibility values in areas closer to the old city—such as Zaoyuan Street, Weiyanggong Street, Beiguan Street, and Daminggong Street—than in the peripheral areas of the city farther away from the old city. Because the spatial distribution of elderly care institutions in Yanta and Weiyang Districts is closer to the old city, with a higher number of elderly care institutions, and with a small difference in population density between the two urban areas, the fundamental reason for the large accessibility difference is the difference in the population accommodated by the elderly care institutions. Compared with Zaoyuan Street and Weiyanggong Street in Weiyang District, the elderly care institutions in Yanta District are more competitive and attractive, so the elderly people are more likely to choose them.

In general, the accessibility values of elderly care institutions in the northwest and northeast are low, while the central city and the south of the city are areas with high accessibility values. The central city has high accessibility values due to its developed economy, convenient transportation, and high-quality elderly care facilities. In addition, the accessibility of elderly care institutions in Textile City Street and Xiwang Street in Baqiao District has higher values compared with the surrounding areas because the clustering of industries leads to a concentrated population, which determines the layout of elderly care institutions in this area. This indicates that a convenient transportation network can help elderly people overcome the effects of distance decay, which would lead to a high elderly care institution capacity per capita. In the northwestern area of Weiyang District, although the density of facilities is high and the population density is low, the value of accessibility is affected by distance decay; thus, the accessibility value is lower in this area. In terms of the overall effect of spatial distribution, the spatial accessibility of elderly care facilities in Xi’an shows a decreasing tendency from the central city and the southern areas to the periphery.

### 4.2. Model Comparison

As shown in Figure 8 and Table 1, the regional accessibility average is 0.1369 when the distance-decay factor is not considered, and the accessibility average is 0.0887 after considering the distance-decay factor. The former service accessibility is 0~3.8495 (Figure 8a), and the latter accessibility range is 0~1.2911 (Figure 8b), compared with the result derived when considering the actual path; therefore, with this consideration, the actual distance between the two places becomes larger, and the spatial accessibility becomes smaller due to the influence of distance decay. In terms of standard deviation, (Figure 8b) is smaller than (Figure 8a), which indicates that the accessibility difference between the six main urban areas of Xi’an is smaller after considering the actual path. Considering the construction of elderly care institutions in relation to the aging of Xi’an in recent years and the selection of the actual path as the distance between two points, (Figure 8b) is closer to the actual situation compared with (Figure 8a).

The value of (Figure 8c) adds a probability function to (Figure 8b): that is, it adds a consideration of the propensity to choose to the travel characteristics of the elderly. In terms of standard deviation, the interval of accessibility values is narrow, and the standard deviation is increased compared with that before adding the probability function, which indicates that the variability among street offices has increased. In terms of the full range, the ranges of accessibility values and standard deviations for the elderly in the two cases are small, indicating that the propensity to choose and the competition for supply have limited impacts on accessibility for the elderly.

## 5. Discussion

The accessibility value of elderly care facilities is affected by many factors; among these factors, distance decay is very important. In this paper, the actual path distance is introduced in terms of distance decay, and the search threshold is designed according to the scale of elderly care institutions and the behavioral characteristics of the elderly. Considering the influence of mutual competitiveness among institutions, the model of the two-step moving search method was improved in this study by adding the selection probability function; the population of the residential area was used as the research unit. Based on GIS spatial analysis technology, the spatial accessibility of elderly care institutions was studied in six urban areas of Xi’an. The improved two-step moving search model can reasonably measure the spatial accessibility of elderly care institutions more scientifically and reliably, and can accurately measure the access of elderly people to elderly care resources in the region, taking into account objective and subjective factors, such as the service radius of elderly care institutions, the number of elderly people in residential areas, and the travel distance required to access these institutions.

### 5.1. Policy Implication 

In Xi’an, where the spatial accessibility of elderly care institutions is unevenly distributed, local government departments should consider their own financial strengths and the spatial rationality of the location of elderly care facilities; accordingly, such policy makers will be able to plan for the necessary number and scale of publicly accessible elderly care institutions within the region. At the same time, based on the industrial advantages and developmental potential of industrial parks, the local government should encourage and guide private elderly care resources to move into the area by means of governmental support and economic subsidies. The research results of this paper can help us to understand the spatial accessibility of elderly care institutions in the main urban area of Xi’an, and can provide references for scientific research, effective planning, reasonable site selection, and the layout of elderly care facilities. The improved model enriches and improves the theory of optimal spatial allocation, while providing a methodological reference for the study of the accessibility of different types of public facilities in other regions. 

### 5.2. Model Shortcomings and Improvements

Due to the many aspects involved in the accurate assessment of spatial accessibility for elderly care institutions, which is difficult to achieve—as well as a series of objective conditions that are not easily overcome—some shortcomings remain in this study. For example, the accuracy of the elderly population data—which were obtained based on the aging rate of each street office multiplied by the corresponding household information—remains insufficient, and a future study will try to analyze its sensitivity; for the choice of the distance-decay function, the Gaussian function, which is currently most commonly used by scholars, was selected. Other forms of decay functions will be discussed in depth in a future study, in conjunction with the characteristics of the study area. With the popularity of bicycle sharing, even the elderly population is not limited to walking, and changes in modes of travel will also affect the threshold setting. In this regard, we will further optimize the existing two-step floating catchment area model by combining the travel characteristics of other travel modes.

Generally, research using the improved 2SFCA model based on GIS has positive theoretical and practical significance for research evaluating spatial accessibility and for planning strategies in elderly care facilities. This paper used the improved 2SFCA method considering the impact of the actual network on the distance-decay calculation and the influence of the behavior of the elderly population on their preferences. The results of the comparison calculations were used to make recommendations for the planning of elderly care facilities. In the future, the project can be improved by further researching the behaviors of the elderly population; this factor was represented by the probability function in this paper.

Kanuganti validated the accuracy and sensitivity of the 2SFCA model by statistically comparing the calculated accessibility values from the E2FCA method with the observed accessibility values [28]. The results of the 2SFCA model and its improved models can be different, but they all exhibited a strong mutual correlation [29].

## 6. Conclusions

This study improved a model for the evaluation of the accessibility of elderly care institutions in the main urban area of Xi’an, and quantitatively evaluated and graphically expressed the accessibility of elderly care institutions with the help of spatial analysis and the visualization functions of GIS. The method adopted in this paper has the following advantages: (1) Considering that elderly people give preferential priority to high service quality and proximity to their homes in decision making, the introduction of the selected probability function means that the model reflects reality more closely, meaning that the results are more accurate; therefore, the layout planning of urban stock space can be better facilitated. (2) The search radius involves distance decay; instead of choosing the plane Euclidean distance, the actual shortest distance along the route is combined with the real road network, thus providing a more accurate calculation result. (3) The service targets of elderly care institutions are members of the elderly population, so the search radius and the distance from the institution to the residential area should be combined at two points with the travel characteristics of the elderly; additionally, in this study, we used the walking speed of the elderly as the travel threshold for the calculation, which is more relevant for these calculations in assessing solutions for the problems that arise with an aging population. (4) With the help of ArcGIS, the results of the evaluation of accessibility were expressed in the form of maps by street offices, which can comprehensively show the spatial distribution characteristics of elderly care facilities. The results of this study can help researchers and decision makers to understand the spatial accessibility of elderly care institutions in the main urban area of Xi’an and can provide a reference for scientific research, effective planning, reasonable site selection, and the layout of elderly care facilities.

## Figures and Tables

**Figure 1 ijerph-19-11465-f001:**
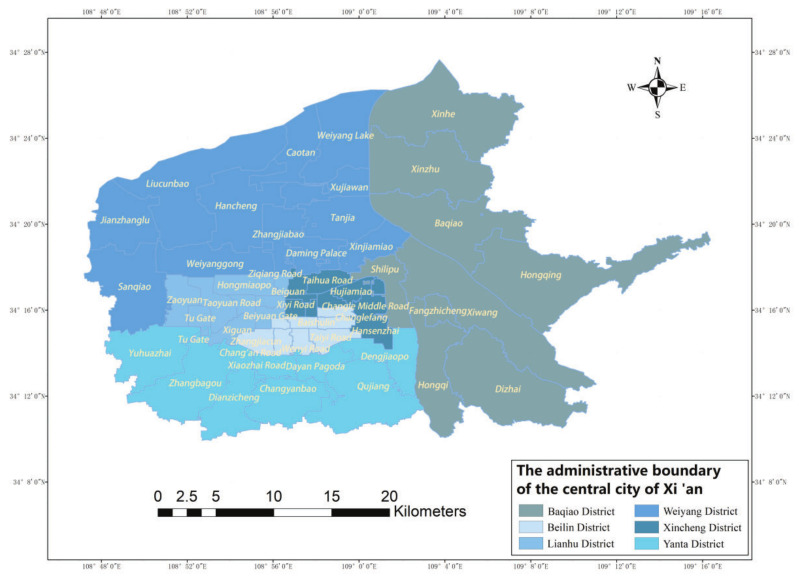
The administrative boundary of the central city of Xi’an.

**Figure 2 ijerph-19-11465-f002:**
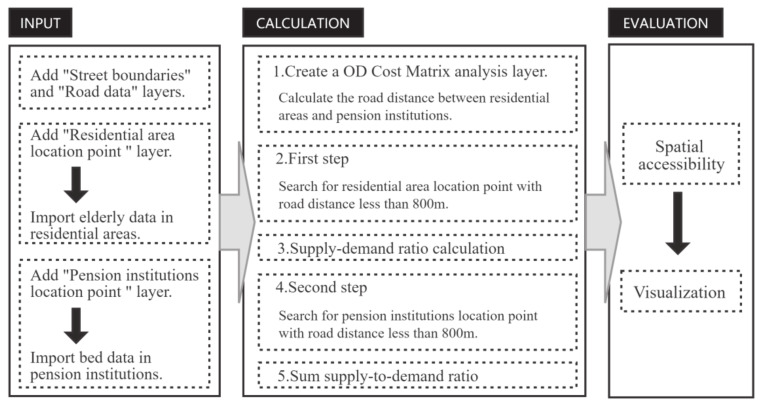
Implementation of the 2SFCA model in the geographic information system (GIS).

**Figure 3 ijerph-19-11465-f003:**
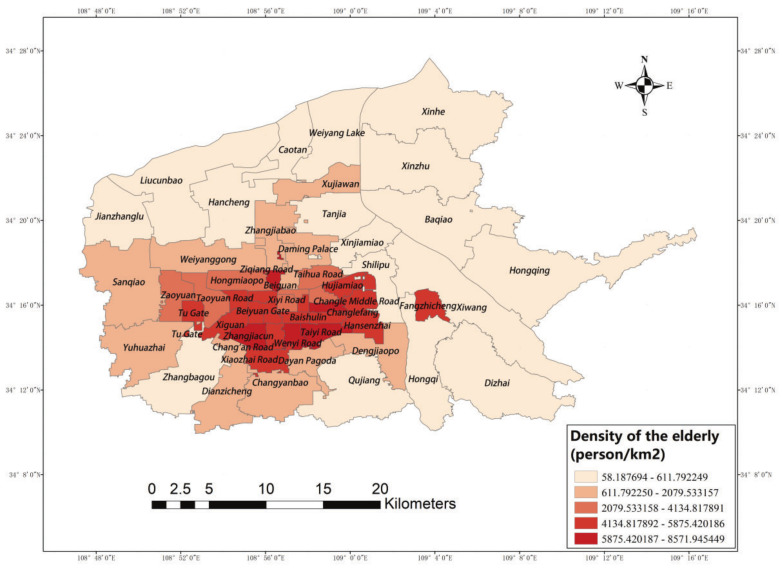
Population density.

**Figure 4 ijerph-19-11465-f004:**
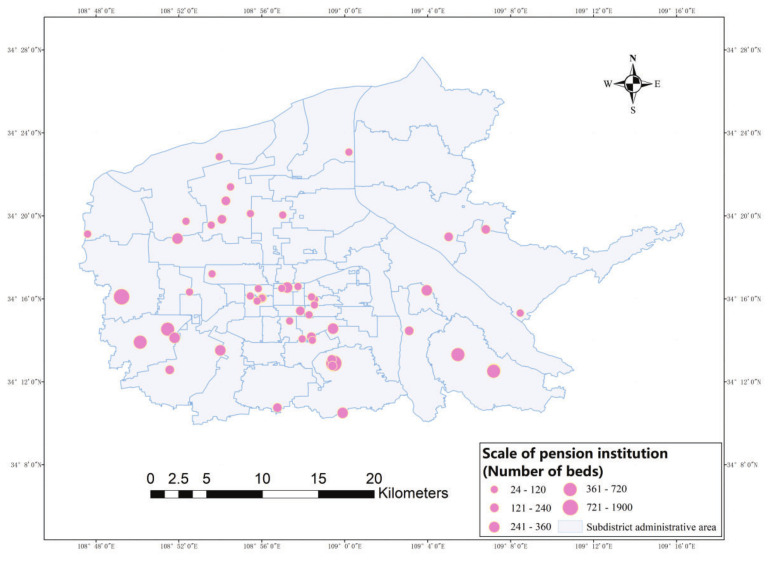
Facility distribution.

**Figure 5 ijerph-19-11465-f005:**
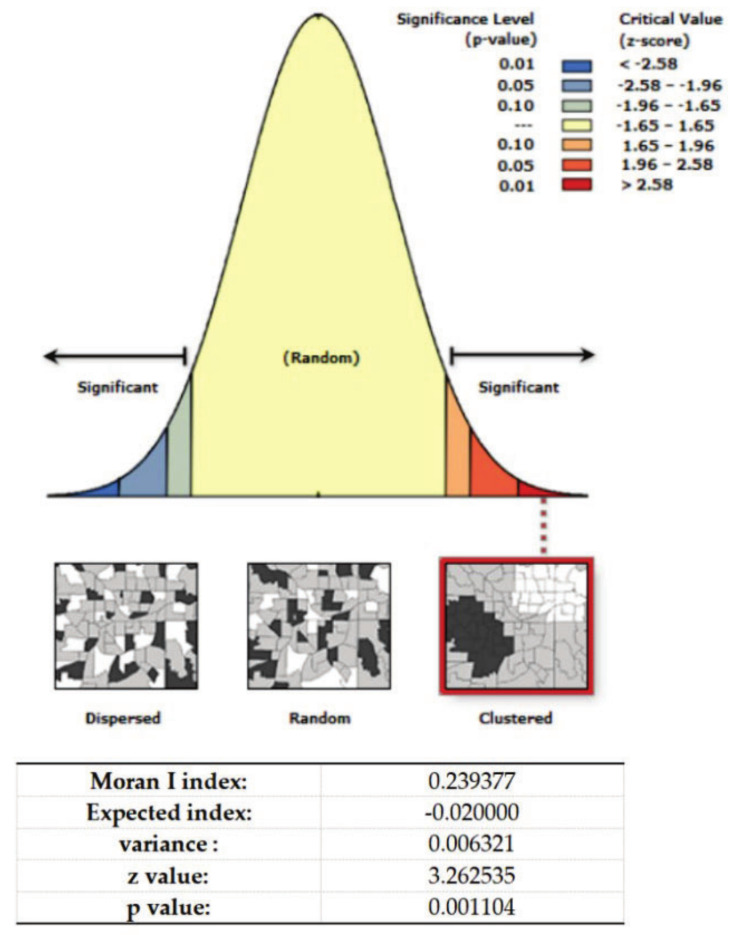
Spatial autocorrelation reports.

**Figure 6 ijerph-19-11465-f006:**
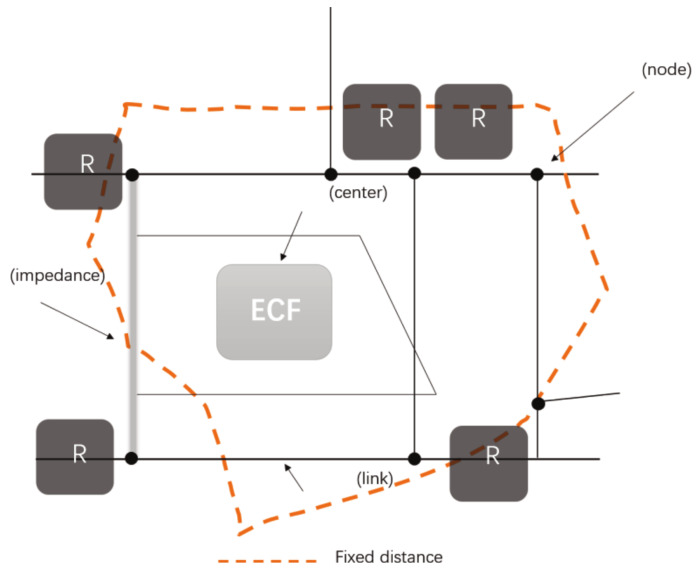
Search radius principle.

**Figure 7 ijerph-19-11465-f007:**
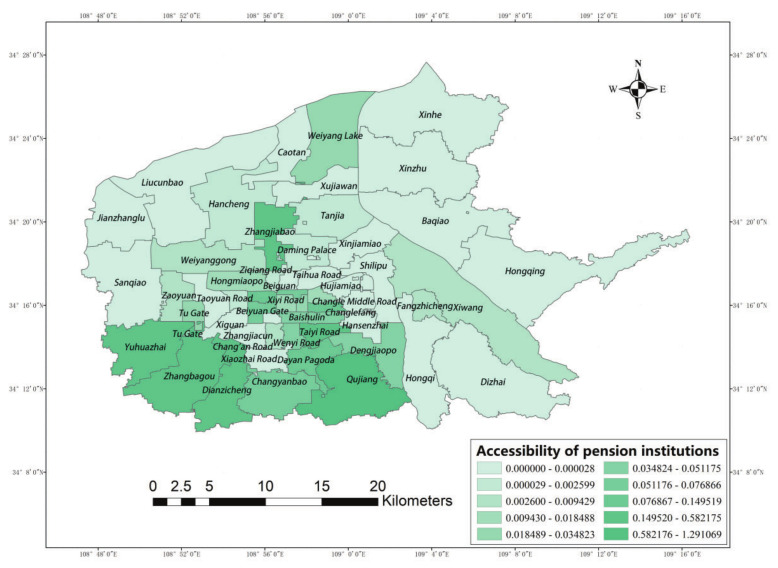
Main urban area accessibility value.

**Figure 8 ijerph-19-11465-f008:**
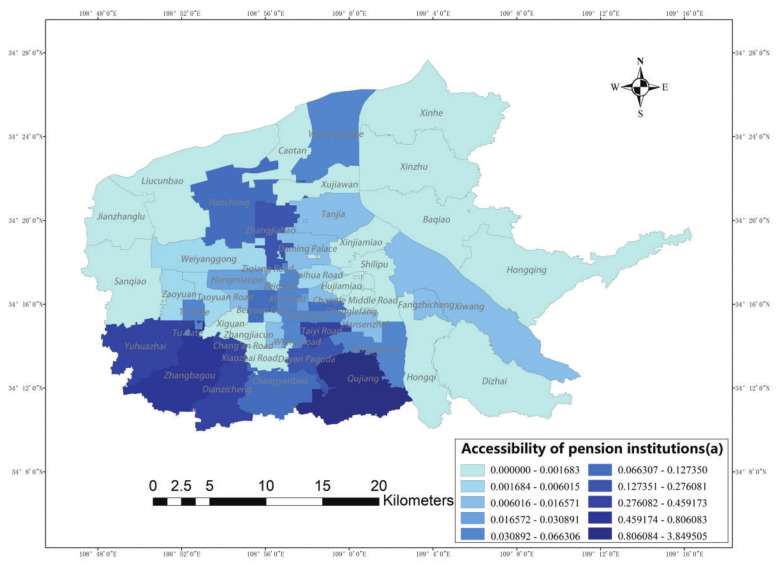
Accessibility of elderly care institutions; (**a**): Before optimization (**b**): Consider the actual path (**c**): Consider the actual path and probability function.

**Table 1 ijerph-19-11465-t001:** Comparison of different models.

Different Models	Average Value	Minimum Value	Maximum Value	Standard Deviation
(a) *A_i_* (Before optimization)	0.136904273	0	3.849505353	0.525043527
(b) *A_i_* (Consider the actual road)	0.088653388	0	1.291069174	0.22078883
(c) *A_i_* (Consider actual roads and probability functions)	0.090737109	0	1.287790313	0.223187781

## Data Availability

Not applicable.

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
