# Peer review of "An Improved Two-Step Floating Catchment Area (2SFCA) Method for Measuring Spatial Accessibility to Elderly Care Facilities in Xi’an, China"

_ijerph, 2022, doi:10.3390/ijerph191811465_

Round 1
Reviewer 1 Report
Thank you for the opportunity to review this manuscript. The authors aimed to improve the model for measuring the accessibility of elderly facilities by introducing a probability function of elderly people's choice in the model, taking into account the influence brought by institutional capacity and service quality. The research idea is well-aligned with aim and scope of the journal. Therefore, I am recommending its publication. I do however have a few minor comments that I believe will enhance the value of the paper.
Xi’an of China should be mentioned in the Abstract.
In literature review of spatial accessibility in background, most of the literature are old. I recommended adding more recent literature.
To increase the significance of the results, the discussion part should embrace the differences and similarities among your findings and those of other scholars.
The conclusion can include the theoretical implications of the research, limitation and future research directions.
The paper was not written in accordance to a typical journal format. Please make it change.
Reviewer 2 Report
This study focuses on improving the mathematical model based on the two-step floating catchment area method (2SFCA) and introduces the probability function of the elderly's choice, considering the influence of institutional capacity and service quality. I have some reservation that can be beneficial to improving the manuscript.
1. Introduction
Line 31 – 42, 64 - 68, 70 - 72: References are missing.
Line 124: This part needs to be discussed in the methodology rather than the introduction section.
The authors need to add the previous studies from developed and developing countries with the applicability of the used model.
The authors need to add the remainder of this paper for the reader's clarity.
2. Data and method
Line 142 - 145: The authors need to add the map that shows administrative divisions and street offices.
Line 152: Why the aging rate is increasing? Please mention the reason(s).
Line 152 - 155: The table might be fine for a clear reader understanding.
Line 160: Please first use the complete form of "POI" and then use the abbreviation for the rest of the article.
Line 160 and 170: OSM is used for Open Source Maps and Open Street Map, respectively. Please correct it.
Figure 1: The authors need to improve the legend as integers.
Figure 3: The authors need to add the significant value at the bottom of the bell shape curve to understand the dispersed, random, and clustered.
Line 265: Reference is missing.
Figure 4: The search radius principle needs to explain more in text.
3. Results
Line 290: Have you discussed "Natural Breakpoint method" in the methodology section?
4. Discussion
The authors need to show the relevance of this study to previous studies and whether this study is consistent or not.
5. Conclusion
Please add the policy implications and practical implications of this study.
Reviewer 3 Report
Re: An improved two-step floating catchment area (2SFCA) method 2 for measuring spatial accessibility to elderly care facilities in 3 Xi’an, China
After carefully reviewing this manuscript, I think this is an interesting paper and this topic about the accessibility measurement of elderly care facilities is worthy of publication. However, its current version needs significant attention in the paper’s structure, methodology, language and English writing. My suggestions to further improve this paper are as follows:
1) Paper structure and contents
i) The general structure of this paper needs to be modified well. Currently, the main body (results, discussion and conclusion) comes up very late while the introduction and methodology sections are already 8 pages (out of 11 pages).
ii) Following the general structure, I expected findings more in the results section. The authors established this paper as a methodological contribution paper, but I had yet to see these points in the results section (if the 2SFCA method works? How it works or even better? Any advancements or validation? Not just about case findings). So I suggest the authors make it clear that it is a big difference between the methodology of this paper and the methodological contribution to theory from this paper. And this can be added with more details in the results section (adding another sub-section?).
iii) In the abstract, the authors missed pointing out the case.
iv) In the conclusion or discussion section, policy implications and suggestions need to be further enhanced.
2) Methodology
i) The authors endeavoured to collect various data for analysis, well done! However, this section now is too long and hard to follow. I suggest authors make a flow diagram to indicate the steps while also summarising the sources of data, to make this methodological process direct and clear.
ii) Line 61-68,124, these sentences are not a general introduction to the method but specific information. So they need to be integrated into the methodological section.
3) Language and English writing
i) The authors need to use a proper academic English writing style. For example, in the introduction section, many references are missing. For example, Line 34, 42, 47, etc., also in P7 L231, references are missing.
ii) Throughout the whole paper, the names of places are too many and confusing. Authors may use acronyms? or numbers? to indicate the places and improve readability.
iii) Engish language needs to be well-checked, both in wording and grammar.
Thanks editors and authors for letting me review this interesting paper, and I wish you good luck.
Reviewer 4 Report
Based on the two-step floating catchment area method, The author improves the mathematical model and contributes to further planning for elder’s care facilities. With the aging society becoming a global issue, the research provides new insights for the planning for elder people. However, the author/s should still consider the followings:
1 The paper’s structure in the current version is not clear enough, for instance, the introduction is too long, and the literature review should be a separate part. The introduction should have more trend of academic argument rather than explain the keywords to reasonable your research question. Similarly, in the literature review part, more academic argument and how this literature support or criticized your current research is needed. In addition, the data source of the method is too long, as far as I am concerned, your focus is on the improved 2SFCA method. I suggest adding one diagram in the methodology part (2.3) to mark the overall step of the research: currently, there are two steps within step 1, while what is the relationship between each step and your improved 2SFCA method? Therefore, the Data source should be concise and methodlogy part should be (2.3) more clear.
2 In Line 257-259, the authors mentioned that the improved model was proposed by Delamater (2013). Although the authors explain it also in the literature part, it is still not clear whether this improved method shows some advantages compared with another method as shown in the literature. This is crucial since you provide an improved method for further research. I have seen it only in the discussion and conclusion, but do not see how it is implicated in the literature and case study part.
3 The result of the research is relatively simple, the author/s detailed analysis of the research findings of the specific urban areas, however, there is a lack of discussion on how the improved 2SFCA model helped to reorganize the elder’s care facilities. As you mentioned in the discussion “The improved two-step moving search model can measure the spatial accessibility of elderly care institutions more scientifically and reasonably….’ The current result could not reflect this. Is that possible to provide a comparison between normal 2SFCA and your improved method? Or you can at least summarize how the new model measures spatial accessibility of the elderly more scientifically and reasonably in the result part?
4 The discussion and conclusion both talk about the advantage of improved 2SFCA, whether this model is applied in China or other countries?
5 Please check carefully about the citation in the text, some of them are ‘(Lee and Kim, 2014) [1] pointed out that… (Line 50)’, (Luo and Wang, 2003; Luo and Qi, 2009; Dai, 2011) [4] (Line 63)’, ‘(Zhou and Matteson, 2016) [2] (Line 54)’ etc. Also, please check your references, there are some mistakes as well. For instance, pp.653-662. (Line 430).
6 Please check all the citation and use the suitable format. In addition, please check the spelling of the formula in the text, for instance, d0 (Line 236), dkj (Line 239), probij(Line 250), f(dik) and others. Please correct if it is ‘Neighbourhood Unit theory’ or ’neighborhood unit theory’, (Line 117). In addition, the English of overall writing should be polished.
Round 2
Reviewer 2 Report
The authors incorporated the all comments.
Reviewer 3 Report
Re: An improved two-step floating catchment area (2SFCA) method 2 for measuring spatial accessibility to elderly care facilities in Xi’an, China
Suggestion: Accept after minor revision
I congratulate the authors that they have much improved the paper quality. Even still, I have one last remark on the introduction section to help finalise this paper for publication. The introduction about this topic needs to be better established, to enhance its logical flow.
In the beginning, “Prominent population growth in urban cities and aging of society are two trends in 33 world’s cities, especially those in developing countries”, I expect more general information worldwide such as developed EU countries, Japan and Korea, and also in developing countries, rather than just directly to the case of China. Then, spatial accessibility to elderly care facilities is important as a global issue but its measurement method is still flawed or not effective enough. Then, you can come up with the proposed method 2SFCA, what and why. At the last of intro, you can introduce the general info of this research and why Xi’an, China can be targeted as the suitable case.
Thank you!
